# Postnatal Overfeeding in Rodents Induces a Neurodevelopment Delay and Anxious-like Behaviour Accompanied by Sex- and Brain-Region-Specific Synaptic and Metabolic Changes

**DOI:** 10.3390/nu15163581

**Published:** 2023-08-15

**Authors:** Andreia Amaro, Diana Sousa, Mariana Sá-Rocha, Marcos Divino Ferreira-Junior, Daniela Rosendo-Silva, Lucas Paulo Jacinto Saavedra, Cátia Barra, Tamaeh Monteiro-Alfredo, Rodrigo Mello Gomes, Paulo Cezar de Freitas Mathias, Filipa I. Baptista, Paulo Matafome

**Affiliations:** 1Coimbra Institute for Clinical and Biomedical Research (iCBR) and Institute of Physiology, Faculty of Medicine, University of Coimbra, 3000-548 Coimbra, Portugal; andreia.amaro15@hotmail.com (A.A.); dianasousa27@hotmail.com (D.S.); mariana.sa.100@gmail.com (M.S.-R.); marcosdfjunior@gmail.com (M.D.F.-J.); daniela.silva26@hotmail.com (D.R.-S.); cat_barra@hotmail.com (C.B.); filipaisabaptista@gmail.com (F.I.B.); 2Center for Innovative Biomedicine and Biotechnology (CIBB), University of Coimbra, 3000-548 Coimbra, Portugal; 3Clinical Academic Center of Coimbra (CACC), 3000-061 Coimbra, Portugal; 4Department of Physiological Sciences, Institute of Biological Sciences, University Federal of Goiás, Goiânia 74690-900, Brazil; gomesrm@ufg.br; 5Laboratory of Secretion Cell Biology, Department of Biotechnology, Genetics and Cell Biology, State University of Maringa, Maringa 87020-900, Brazil; saavedralpj@gmail.com (L.P.J.S.); pmathias@uem.br (P.C.d.F.M.); 6Internal Medicine Department, University Hospital Center of Coimbra, 3004-561 Coimbra, Portugal; 7Coimbra Health School (EsTeSC), Polytechnic University of Coimbra, 3046-854 Coimbra, Portugal

**Keywords:** lactation, overfeeding, neurodevelopment, anxious-like behaviour, energy balance, synaptic balance

## Abstract

Nutritional disturbances during the early postnatal period can have long-lasting effects on neurodevelopment and may be related to behavioural changes at adulthood. While such neuronal connection disruption can contribute to social and behaviour alterations, the dysregulation of the neuroendocrine pathways involved in nutrient-sensing balance may also cause such impairments, although the underlying mechanisms are still unclear. We aimed to evaluate sex-specific neurodevelopmental and behavioural changes upon postnatal overfeeding and determine the potential underpinning mechanisms at the central nervous system level, with a focus on the interconnection between synaptic and neuroendocrine molecular alterations. At postnatal day 3 (PND3) litters were culled to three animals (small litter procedure). Neurodevelopmental tests were conducted at infancy, whereas behavioural tests to assess locomotion, anxiety, and memory were performed at adolescence, together with molecular analysis of the hippocampus, hypothalamus, and prefrontal cortex. At infancy, females presented impaired acquisition of an auditory response, eye opening, olfactory discrimination, and vestibular system development, suggesting that female offspring neurodevelopment/maturation was deeply affected. Male offspring presented a transitory delay in locomotor performance., while both offspring had lower upper limb strength. At adolescence, both sexes presented anxious-like behaviour without alterations in short-term memory retention. Both males and females presented lower NPY1R levels in a region-specific manner. Furthermore, both sexes presented synaptic changes in the hippocampus (lower GABA_A_ in females and higher GABA_A_ levels in males), while, in the prefrontal cortex, similar higher GABA_A_ receptor levels were observed. At the hypothalamus, females presented synaptic changes, namely higher vGLUT1 and PSD95 levels. Thus, we demonstrate that postnatal overfeeding modulates offspring behaviour and dysregulates nutrient-sensing mechanisms such as NPY and GABA in a sex- and brain-region-specific manner.

## 1. Introduction

The early postnatal period is critical for brain maturation and growth, being maternal nutritional cues determinants for milk quality and quantity and consequently for newborn health [1]. Nutritional disturbances during this period can alter the supply of nutrients to the offspring and consequently have long-lasting effects on theirown growth and metabolism [1]. Besides genetic factors, non-genetic factors may also evoke structural and functional changes in neuronal activity and metabolism, with impacts on neurodevelopment and behaviour throughout life [1,2]. Overfeeding is often a consequence of maternal obesity and the consumption of unbalanced diets is one of the factors that can contribute to disturbances in neuronal circuits during early life, namely the proper communication between central and peripheral structures [3,4,5]. Neonatal overfeeding models have been studied as representative of childhood obesity. In the small litter model, both male and female pups are raised with greater access to their mother’s milk and, as a consequence, have higher susceptibility to the onset of metabolic diseases such as adiposity, insulin resistance, obesity, and type 2 diabetes, featuring a metabolic syndrome phenotype [3,6,7]. Besides these consequences, postnatal overfeeding may also contribute to reshaping neuronal connections and increase the susceptibility to behavioural alterations such as anxiety, deficits in sociability, memory, and learning impairments [8,9].

In the central nervous system (CNS) architecture, both the hippocampus and prefrontal cortex play an important role in the regulation of emotions, cognition, and motivation, being the central targets for neuroendocrine signals. The establishment of synapses and neuronal network formation are crucial during development, being GABA an important neurotransmitter involved in the inhibitory/excitatory balance [10,11,12]. Studies have demonstrated that the manipulation of the GABAergic system during the critical developmental window is implicated in emotional and neuropsychiatric disturbances such as depression and anxiety-like behaviours, pointing out thatGABA imbalance may contribute to behavioural impairments [12,13,14,15].

In the CNS, the hypothalamus is fundamental in sensing and controlling energy expenditure, being the arcuate nucleus (ARC) region, in particular, the master regulator of whole-body energy homeostasis [16,17]. Metabolic hormones, such as ghrelin and insulin, integrate hormonal and nutritional metabolic signals from the peripheral circulation that can influence hypothalamus development [16,17,18]. The metabolic regulation of the hypothalamus is orchestrated by two specific types of neurons, the orexigenic (appetite-stimulating) neuropeptide Y (NPY) and agouti-related peptide neurons that directly stimulate food intake, and by the anorexigenic (appetite-suppressing) pro-opiomelanocortin (POMC) neurons. Furthermore, in the hypothalamus, NPY neurons release GABA to directly inhibit POMC neurons [16,17]. Therefore, a dysregulation of this system can compromise energy balance homeostasis and contribute to the onset of metabolic disease progression, including obesity and type 2 diabetes [19,20].

Besides its function in homeostatic feeding, the NPY system also plays an important modulatory role in learning and memory, cognition, anxiety, and neuroplasticity [21]. Studies have demonstrated that NPY1R in the prefrontal cortex, hippocampus, and hypothalamus plays an important role in the regulation of anxiety-related behaviours [13,21,22,23,24]. Although the mechanisms behind these alterations are currently poorly understood, the interplay between the hippocampus and the prefrontal cortex with the hypothalamus seems to be important in regulating feeding, energy balance, and behaviour.

Taking this into consideration, this study aims evaluate the impact of postnatal overfeeding on offspring neurodevelopmental and behaviour, according to sex -specificities, elucidating the putative molecular alterations that may be underlying them. Furthermore, we intended to understand the molecular key players that contribute to behavioural and nutrient-sensing abnormalities upon postnatal overfeeding conditions.

We hypothesize that offspring overweight impairs the neurotransmitter/synaptic balance and nutrient-sensing mechanisms, inducing neurodevelopmental and behavioural changes at the adolescent period.

## 2. Materials and Methods

### 2.1. Animal Maintenance and Experimental Design

The study was performed at the Coimbra Institute for Clinical and Biomedical Research (iCBR), at the Faculty of Medicine of the University of Coimbra, and performed according to good practices of animal handling, namely with the approval of the Institutional Animal Care and Use Committee (ORBEA 13/18). All procedures were performed by licensed users by the Federation of Laboratory Animal Science Associations (FELASA), according to the European Community directive guidelines (2010/63/EU) for the use of experimental animals.

Pregnant Wistar rats were housed under standard animal conditions—1 animal per cage—with a temperature of 22–24 °C, 50–60% humidity, a standard light cycle (12 h light/12 h darkness), and ad libitium access to water and food (standard diet 4RF21). After offspring birth, all litters were culled to 8 pups per dam for normalisation. To study the effects of postnatal overfeeding and overweight, a small litter (SL) procedure was considered where, at postnatal day (PND) 3, the number of pups per dam was reduced from 8 to 3. Both male and female offspring were subjected to a small litter protocol, in which each dam had three pups of the same sex (Figure 1A).

During the breastfeeding period, dams’ body weight was weekly motorized. On the weaning day—PND21—both male and female offspring were separated from the progenitor and kept 3 animals per cage were kept until PND45, with ad libitum access to water and food (standard diet 4RF21). At the adolescence period, PND45, animals were anesthetised by intraperitoneal (IP) injection of ketamine/chlorpromazine and euthanised by cervical displacement, followed by the collection of the hippocampus, hypothalamus, and prefrontal cortex and storage at −80 °C for further molecular analysis (Figure 1A).

### 2.2. Maternal Milk Collection and Composition Analysis

At PND21, female dams were injected via IP with oxytocin (Facilpart) with a concentration of 10 UI/mL, after 6 h of fasting, followed by milk sample collection and storage at −80 °C. For milk composition analysis, triglycerides were measured, and total antioxidant capacity levels were determined following the Assay Kit (ab65329 Abcam) manufacturer’s instructions.

### 2.3. Infancy Neurodevelopmental Tests

During the infancy period, female and male offspring from the control and SL groups were tested in several developmental tests from PND5 to PND17. The following neurodevelopment tests were monitored each day during the light phase of the light cycle and under dim white light.

#### 2.3.1. Cliff Aversion

During PND6 to PND10, pup’s vestibular system development was evaluated, as an indicator of balance and coordination. Each animal was placed on a flat raised surface with its snout and front paws hanging over the edge, and the time that each pup took to move its paws and snout away from the edge was recorded up to 30 s.

#### 2.3.2. Nest Seeking Behaviour

From PND5 until PND10, pup’s ability to discriminate its nest bedding was determined using a rectangular arena (26 × 14 cm) divided into 3 compartments: the center compartment where the pup was placed, the home bedding goal on one side (nest bedding), and a fresh bedding goal on the opposite side (fresh clean bedding). In the first trial, each animal was placed in the central compartment from the goal compartment, and for the second trial, the animal was facing the opposite side and tested to avoid possible preferences. The latency to the goal was scored as the time that the animal took to transpose the apparatus home bedding goal mark with both snout and forelimbs. The cut-off trial was 120 s and latency to the goal was estimated as an average between the two trials, per day. 

#### 2.3.3. Wire Suspension Test

To assess the pup’s forelimb strength, they were placed against a horizontal wire rod, allowed to grasp it with both forepaws, and the time was scored up to 10 s after the release of the animal.

#### 2.3.4. Locomotion

From PND5 to PND14, pup’s locomotor ability was assessed by placing the animals in the center of a 13 cmdiameter circular flat arena. The time that each animal took to fully exit the arena with all four limbs was recorded up to 30 s. 

#### 2.3.5. Auditory Startle

From PND11 until PND14, the auditory startle response was evaluated to measure auditory system development, as an indicator of somatosensory, vestibular, and/or proprioceptive maturation and function. Animals’ capacity to produce a full-body startle response to a loud finger snap was evaluated and the percentage of animals that responded, per litter, per day, was calculated. 

#### 2.3.6. Eye Opening

From PND13 until PND17, the eye opening of each animal was monitored and the percentage of pups with eyes opened, per litter, per day, was calculated.

### 2.4. Middle Adolescence Behavioural Tests

All animals were habituated to the experimentation room under dimmed red light, a controlled temperature (22–24 °C), and ventilation, and behavioral tests were performed during the light phase of the light cycle. 

#### 2.4.1. Open Field Test (OPF)

At PND43, the OPF was performed to assess locomotor ability and explorative behaviour. Animals were placed in the center of the arena (45 × 45 × 40 cm) and were left to freely explore it for 10 min. Test analysis were performed through the ANY Maze software to evaluate the animal’s locomotor pattern of exploration, average speed, distance traveled and time spent in the arena center and periphery.

#### 2.4.2. Elevated plus Maze Test (EPM)

At PND43, anxious-like behaviour was assessed, where the animals freely explored for 5 min the open and closed arms of the maze. Through the analysis of the *Observador* software, it was calculated the ratio of time spent in the open arms/total time spent in both open and closed arms, which was used as an index of anxious-like behaviour. Additionally, frequencies of entries was also evaluated.

#### 2.4.3. Novel Object Recognition Test (NOR)

At PND44, the NOR test was performed to evaluate short-term recognition memory, through the discrimination of familiar and novel objects. The test was divided into 2 phases, a familiarisation trial where the animals explored, for 10 min, 2 objects equal in their shape, colour, and height, and, after a 4 h inter-trial interval, a test trial where the animals explored, for 3 min, 2 objects with equal height but a different shape and colour. Animal performance was measured through the Recognition Index (RI): [RI = TN/(TN + TF)] and Discrimination Index (DI): [DI = (TN − TF)/(TN + TF), where TN is the time spent exploring the novel object and TF the time exploring the familiar one.

### 2.5. Western Blot

At PND45, the hippocampus, hypothalamus, and prefrontal cortex were collected and stored at −80 °C. For Western blot analysis, tissues were homogenised in lysis buffer (0.25 M Tris-HCl, 125 mM NaCl, 1% TritonX-100, 1 mM EGYA, 1 mM EDTA, 20 mM NaF, 2 mM Na_3_VO_4,_ 10 mM β-glycerophosphate, 2.5 mM sodium pyrophosphate, 10 mM PMSF, 40 µL of protease inhibitor) using the TissueLyser system for 10 min. The bicinchoninic acid (BCA) protein assay kit was carried out on the supernatant (obtained by centrifugation at 14.000 rpm for 20 min at 4 °C). After the addition of Laemmli buffer (62.5 mM Tris-HCl, 10% glycerol, 2% SDS, 5% β-mercaptoethanol, and 0.01% bromophenol blue) and denaturation, tissue samples (15 µg) were loaded onto SDS-PAGE and electroblotted into a polyvinylidene difluoride (PVDF) membrane. Tris-buffered saline-tween (TBS-T) 0.01% and bovine serum albumin (BSA) 5% were used to block the membranes, which were then incubated with primary (overnight, 4 °C) and secondary antibodies (2 h, room temperature), as described in Table 1.

Immunoblots were detected using enhanced chemiluminescence (ECL) substrates and the LAS 500 System. The bands were quantified with the Image Quant^®^ software, and the results were expressed as a percentage of the control and normalised to the loading control (calnexin, 83 kDa or GADPH, 37 kDa).

### 2.6. Statistical Analysis

The results are presented as the mean ± standard error of the mean (SEM). In the neurodevelopmental and behavioural tests, we considered 9 to 20 males per group and 6 to 12 females per group. For the Western blot analysis, it was considered 6 male animals for both experimental groups, 5 female animals in the control group, and 4 female animals in the SL group. Each lane represented a different animal per group. The normality of the data was assessed with the Shapiro–Wilk normality test. Student’s *t*-test was used to determine the differences between the control and SL groups. Statistical analysis was performed with GraphPad Prism 9 and values of *p* < 0.05 were considered significant.

## 3. Results

### 3.1. Postnatal Overfeeding Modifies Milk Composition and Quality

After birth, pups from the control and SL experimental groups were maintained with the dams until the weaning day (PND21). Dams’ body weight composition was weekly monitored, demonstrating no alterations during the lactation period (Figure 1B). This period is crucial for a newborn’s healthy growth since it is exclusively fed by maternal milk. Therefore, changes in nutritional milk composition can result in lifelong changes in the offspring metabolic profile, contributing to the onset of obesity and metabolic syndrome. At the weaning day, milk composition and quality content from SL dams was analysed, showing decreased total antioxidant capacity (*p* < 0.05) and triglyceride levels (*p* < 0.01) compared with the control group (Figure 1C,D).

Regarding pups’ body weight, only the male offspring demonstrated a higher body weight at weaning and PND45. Females did not show any differences in body weight, although they had other metabolic disturbances such as hypoinsulinemia and hypertriglyceridemia [25].

### 3.2. Postnatal Overfeeding Induces a Sex-Dependent Neurodevelopmental Delay

Both female and male pups were tested in several developmental tests during the first days of life to assess the effects of postnatal overfeeding on offspring neurodevelopment.

Pups were tested in nest-seeking test, to determine pup’s olfactory ability to discriminate home bedding and as an indicator of maternal bond.

Male animals did not show alterations compared to the control group (Figure 2B), or in tests dependent on their vestibular system maturation (Figure 2A). At PND13, a transitory impairment (*p* < 0.05) in locomotor ability was noted, as well as less strength (*p* < 0.01) in the upper limbs (Figure 2C,D). Nevertheless, no alterations were observed in the auditory startle response and eye-opening day, demonstrating no alterations in the acquisition of these developmental milestones (Figure 2E,F).

Regarding female pups, postnatal overfeeding during breastfeeding induced a significant delay in the maturation of the vestibular system at PND7 (*p* < 0.05) and PND8 (*p* < 0.01), as assessed in the cliff aversion test (Figure 2G). A similar delay was also observed in the maturation of the olfactory system or impaired maternal attachment at PND6 (*p* < 0.05) and PND7 (*p* < 0.01) (Figure 2H). Although no major alterations in locomotion were observed, females presented significantly less strength in their upper limbs at PND12 (*p* < 0.05) (Figure 2I,J). Furthermore, females presented a significant delay in the auditory startle response at PND12 (*p* < 0.01) and in the eye-opening day at PND14 (*p* < 0.01) and PND15 (*p* < 0.05) (Figure 2K,L), suggestive of a neurodevelopmental delay, an effect not observed in males.

### 3.3. Postnatal Overfeed Display an Anxious-like Behaviour in Both Males and Females

Few studies have addressed the behavioural consequences of postnatal overfeeding/overweight during lactation. During the breastfeeding period, postnatal overfeeding induced a delay in neurodevelopment in a sex-dependent manner. We aimed to understand whether those alterations could be associated with behavioural alterations in the adolescence phase.

Overweight male offspring group spent less time in the open arms (*p* < 0.05) in the EPM test and entered less frequently in the open arms (*p* < 0.05). On the opposite, a trend to increased entry frequency in the closed arms (*p* = 0.0566) was observed compared with the control group (Figure 3A–C), suggesting an anxious-like behaviour. We also evaluated the locomotor activity and explorative behaviour in the OPF test, analysing the time that each animal spent exploring the center and the periphery of the OPF arena. SL males spent similar time exploring the center of the arena as the controls (Figure 3D), and no changes were found concerning the distance travelled and mean speed, demonstrating no alterations in locomotor activity (Figure 3E,F). Additionally, both control and SL male groups presented similar recognition (RI) and discrimination indexes (DI), demonstrating no significant alterations in short-term memory processes (Figure 3G,H).

Similarly, to males, females also spent less time exploring the open arms in the EPM test (*p* < 0.05), together with a higher entry frequency in the closed arms (*p* < 0.05) (Figure 3I–K). In accordance, females also spent less time exploring the center of the OPF arena (*p* < 0.05), further indicating an anxious-like behaviour (Figure 3L). These alterations were not influenced by impairments in locomotor ability, since no changes were detected in the total distance travelled and mean speed in the OPF test (Figure 3M,N). Also, no significant alterations were found regarding female short-term recognition memory, since no alterations were observed in the time exploring the familiar and novel objects (Figure 3O,P).

### 3.4. Sex-and Brain-Region-Specific Alterations in NPY System upon Overfeeding Conditions

Anxiety and stress-related diseases may be related to nutrient-sensing and synaptic changes dependent on food availability. Therefore, we evaluated whether postnatal overfeeding may be involved in nutrient-sensing dysregulation and behaviour alterations, namely in the NPY machinery and ghrelin receptor—GHS-R1α—since it is involved in food intake stimulation.

In the hippocampus of male overweight, no alterations were observed in NPY1R, NPY2R, and GHS-R1α protein levels (Figure 4A–C), whereas, in female offspring, it was observed a significant decrease in NPY1R (*p* < 0.001) and GHS-R1α (*p* < 0.05) protein levels (Figure 4D,F), without alterations in NPY2R levels (Figure 4E).

In the prefrontal cortex, both male and female SL offspring presented lower NPY1R levels (*p* < 0.05 and *p* < 0.01, respectively) (Figure 4G,J). Only male offspring from the SL group presented significantly lower NPY2R (*p* < 0.01) and GHS-R1α (*p* < 0.01) levels (Figure 4H,I), without alterations in females (Figure 4K,L).

Concerning the hypothalamus region, where the NPY pathway is involved in regulating homeostatic feeding, both males and females from the SL group had significantly lower NPY1R (*p* < 0.01 and *p* < 0.05, respectively) protein levels (Figure 4M,P), without alterations in NPY2R and GHS-R1α levels (Figure 4N,O,Q,R).

### 3.5. Early Overfeeding Induces a Sex- and Brain-Region-Specific Synaptic Imbalance

During the perinatal period and until late adolescence, important key events ensure the proper development and maturation of the CNS, including neuronal network formation, which comprises the formation of excitatory and inhibitory synaptic connections. Taking this into consideration, it is important to evaluate whether early exposure to excessive nutrients may compromise the proper formation of synapses in brain regions involved in homeostatic feeding and behaviour.

In the hippocampus of males from the SL group, no significant alterations were observed in proteins related to the glutamatergic synapse, namely glutaminase C, an enzyme responsible for converting glutamine into glutamate (Figure 5A), and vesicular glutamate transporter (vGLUT1) protein levels (a marker of glutamatergic synapses, involved in the transportation of glutamate into the synaptic vesicles), and postsynaptic density protein (PSD)-95 levels, suggesting that postnatal overfeeding does not induce alterations at excitatory synaptic level (Figure 5B,C). Regarding the inhibitory synapse, at the presynaptic site, no alterations were detected in vesicular GABA transporter (vGAT) protein levels (involved in the transportation of GABA into synaptic vesicles at the presynaptic site and a marker of GABAergic synapses) (Figure 5D). However, significantly higher GABA_A_ receptor protein levels (*p* < 0.01) were noted, without alterations in the postsynaptic scaffolding protein—gephyrin (Figure 5E,F).

Female offspring from the SL group also showed a trend to higher glutaminase C protein levels (*p* = 0.0669) in the hippocampus, although without alterations in vGLUT1 and PSD95 protein levels (Figure 5G–I). In the GABAergic synapse, significantly lower vGAT (*p* < 0.05) and GABA_A_ (*p* < 0.01) receptor protein levels were noted, without alterations in gephyrin protein levels (Figure 5J–L).

In the male prefrontal cortex from in the SL group, higher glutaminase C (*p* < 0.01) levels were noted without alterations in vGLUT1 and PSD95 protein levels, similar to what was observed in the hippocampus (Figure 6A–C). At the presynaptic site, no alterations were observed in vGAT protein levels, although GABA_A_ receptor protein levels were shown to be augmented (*p* < 0.001) (Figure 6D,E). Interestingly, at the post-synaptic site, significantly lower levels of the scaffold inhibitory protein gephyrin (*p* < 0.001) were noted in the SL group compared with the control group (Figure 6F).

Regarding females, no significant alterations were observed in proteins involved in the glutamatergic synapse, namely glutaminase C, vGLUT1, and PSD95 (Figure 6G–I). Similar to vGLUT1, no significant changes were detected in vGAT protein levels (Figure 6J). However, at the inhibitory post-synaptic site, similar to male offspring, higher GABA_A_ receptor levels (*p* < 0.05) and lower gephyrin levels were noted (*p* < 0.01), suggesting that postnatal overfeeding induces a possible analogous synaptic imbalance in this region in both sexes (Figure 6K,L).

In the male hypothalamus from SL group, no major differences were found at glutamatergic synapse levels, namely regarding vGLUT1 and PSD95 protein levels (Figure 7A,B). Similarly, at the GABAergic synapse, no significant alterations were observed at the pre-synaptic level, through vGAT protein levels, although GABA_A_ receptor presented a trend to lower levels (*p* = 0.0781) (Figure 7C,D).

Concerning female offspring hypothalamus, significantly higher levels of glutamatergic proteins, vGLUT1 (*p* < 0.05) and PSD95 (*p* < 0.05), were noted at the pre- and postsynaptic sites, respectively (Figure 7E,F). Interestingly, vGAT and GABA_A_ receptor proteins involved in the inhibitory synapse showed no major differences, suggesting that postnatal overfeeding may induce a synaptic imbalance in the female hypothalamus (Figure 7G,H).

### 3.6. Postnatal Overfeeding Modulates AMPK Signalling in a Sex- and Brain-Region-Specific Manner

As mentioned before, postnatal overfeeding during the breastfeeding period induced a higher body weight gain in male offspring that was maintained until the adolescence period (Figure 8A). At PND45, lower plasma insulin levels and increased triglyceride and high-density lipoprotein (HDL) levels were also observed, as previously published (Figure 8A) [25]. However, in females, no significant alterations were observed in body weight gain during and after breastfeeding, and, similarly to males, lower plasma insulin levels and increased HDL and cholesterol levels were noted, as previously reported (Figure 8B) [25].

Since metabolic alterations were observed in both male and female offspring, especially in plasma insulin levels, we hypothesized that insulin signalling impairment in the hippocampus and in the prefrontal cortex could be involved in the neurodevelopmental and behavioural alterations detected. In the male hippocampus from the SL group, no alterations were observed in total insulin receptor (IRtotal) and its phosphorylation—IRp (Figure 8C,D). In addition to that, the total and phosphorylated levels of AMP-activated protein kinase (AMPK) remained unaltered (Figure 8E,F). In the male prefrontal cortex, lower IRtotal levels were observed (*p* < 0.05), without alterations in its phosphorylation (Figure 8G,H). Furthermore, lower total and phosphorylated AMPK protein levels were noted (*p* < 0.001 and *p* < 0.001, respectively) (Figure 8I,J).

Females showed no significant alterations regarding total and phosphorylated insulin protein levels in both hippocampus and prefrontal cortex regions (Figure 8K,L,O,P). In the hippocampus region, higher AMPK phosphorylated levels (*p* < 0.01) were noted, although lower total AMPK (*p* < 0.05) levels were observed, suggesting that the higher activation could be a compensatory mechanism due to the lower availability of AMPK (Figure 8M, N). Additionally, in the prefrontal cortex region, lower total AMPK levels were also noted (*p* < 0.05), despite no alterations observed in its activation (Figure 8Q,R).

## 4. Discussion

During the perinatal period, maternal cues may have a deeply impact on offspring growth and development, being determinant for metabolic and neuronal circuits shaping and maturation. After birth, offspring is exclusively nourished by maternal milk, supporting the importance of maternal healthy maternal nutrition and habits during this period [1]. In this work, we studied the consequences of postnatal overfeeding on offspring regarding their neurodevelopment, behaviour, and putative synaptic and nutrient-sensing alterations in different brain regions involved in feeding, emotion, and cognition. Postnatal overfeeding is used as an experimental model of childhood obesity that is sustained throughout adulthood and is characterized by a reduced number of pups, during the first days of life (from 8 to 3 pups), which results in higher milk availability and consequently increased milk and caloric intake by the offspring [3,4].

In the presented study, litter reduction procedure induced a significant reduction in total antioxidant capacity and triglyceride levels in milk. In overfeeding conditions, a significant reduction in antioxidant machinery function was already described, namely in glutathione reductase and peroxidase, superoxide dismutase, and catalase activities which can contribute to lower total antioxidant capacity levels [26]. Furthermore, after litter reduction, there is a hyperfunction of prolactin that continuously stimulates the production of milk from the mammary gland, which can consequently affect milk composition. In rodent models of food restriction during lactation, there is a significant reduction in milk production, due to lower prolactin levels [27]. In line with that, prolactin inhibition, at the end of lactation, induces less milk production, accompanied by higher triglyceride levels [28].

In our previous work, we demonstrated that male and female offspring have alterations regarding their metabolic profile, namely lower plasma insulin and higher HDL levels, although only males developed an overweight phenotype [25]. Herein, we found lower total insulin receptor levels in the male prefrontal cortex without alterations in its phosphorylation, together with lower total and phosphorylated AMPK levels, possibly as a direct response to lower plasma insulin levels. Females also presented lower AMPK levels in both regions, although, in the hippocampus, higher levels of AMPK phosphorylation were noted.

Our aim was to evaluate the influence of postnatal overfeeding in the achievement of important developmental milestones. Offspring males from the SL group did not present significant alterations regarding vestibular system maturation, olfactory discrimination, auditory ability, and eye-opening day. Only a transitory impairment was noted in their locomotor ability and strength, which could be an effect of their higher body weight gain. On the opposite, female offspring under postnatal overfeeding conditions have a deep delay in its milestone achievements, namely in vestibular system maturation, impaired strength, olfactory discrimination, and auditory ability and a delay in eye-opening day, suggesting that females are more susceptible to early postnatal overfeeding than males. To our knowledge, this is the first study addressing the impact of postnatal overweight on offspring neurodevelopment and considering sex-specificities. Indeed, only one study, performed by Novais et al., demonstrate that males from the SL group did not have significant alterations in their neurodevelopment achievements, namely olfactory discrimination and locomotor activity, which corroborates our findings [29].

Postnatal overfeeding affects offspring behaviour throughout life, leading to an anxious-like phenotype at adolescence. Some studies are in accordance with our findings, demonstrating that neonatal overfeeding predisposes to anxiety or more stress-related behaviour [5,8]. However, some studies have addressed that at early infancy (PND14), males do not show alterations in their behaviour, whereas females present enhanced exploratory behaviour and reduced anxiety in the elevated plus maze test [30]. In this regard, offspring behaviour may be influenced by maternal care. To corroborate this, a study performed by Enes-Marques et al. demonstrate that maternal care during the lactation period increases pup licking time by mothers and decreases the time that the mother spent away from the offspring. As a result, due to increase maternal care, SL offspring have less anxious and explorative behaviour in adulthood, suggesting that maternal care is crucial in shaping offspring behaviour during life [9,31].

From a social perspective, anxious-like behaviour may be viewed as an adaptative response to an unknown environment, with animals preferring to be isolated rather than searching or exploring for food. In accordance with this, females from the SL group show less explorative behaviour in the OPF test, while males show no significant alterations, suggesting that females may have less motivation to explore than males. Interestingly, in a model of maternal separation, where male mice experience hunger and improper rearing conditions, during the early life period, there is an adaptative behaviour that induces increased foraging and reduced competition within the social group, and, as a result, they display a long-lasting submissive phenotype, with increased social recognition and explorative behaviour [13]. Furthermore, maternal undernutrition, which results in offspring malnourishment conditions during the postnatal period, also displays a similar pattern due to increasing explorative and reduced anxiety and stress-related behaviours in the offspring, suggesting that, in under- or overnutrition conditions, there are adaptive responses that modulate anxious-like behaviour [32,33]. On the other hand, maternal high-fat diet consumption during the perinatal period has been associated with an increase in anxiety-like behaviour in the offspring, suggesting that maternal cues can also trigger a direct role in offspring behaviour [24,34,35,36,37,38,39]. Besides stress-related behaviour alterations, the postnatal overfeeding model is associated with an increased risk of autism-like patterns, namely a decrease in social interaction, deficits in social communication, and repetitive and stereotyped behaviours [29].

Studies have demonstrated that neonatally overfed rats have a hyperreactive hypothalamus-pituitary-adrenal (HPA) axis, accompanied by elevated stress-induced corticosterone levels in adulthood, suggesting that these modifications may play a role in anxiety-related behaviour [8,30,38]. NPY1R plays a significant role in the regulation of the HPA axis [40]. However, little is known about the role of NPY signalling in mediating metabolic and neuronal changes.

Limbic neuropeptide Y1R may be a possible mechanism related to postnatal metabolic and behavioural changes. Studies have demonstrated that during weaning, male offspring from the SL group presented an increased expression of POMC, NPY, and ghrelin (GHS-R1α) receptor in the ARC hypothalamic region, which contributes to satiety control dysregulation [40,41]. On the other hand, NPY1R has a strong effect in suppressing neurons located in the ventromedial nucleus of the hypothalamus, suggesting that NPY1R may have a different role according to its hypothalamic region [5,42]. In adulthood, females exhibit higher NPY and lower POMC contents in the ARC, impacting their mechanisms of satiety control [43]. Our results shows that adolescent animals of both sexes have lower NPY1R levels in the hypothalamus, suggesting that, under postnatal overfeeding conditions, there is a continuous hypothalamic remodelling that results in a decrease in food intake stimulation, which shifts at adulthood, contributing to the onset of metabolic dysfunction. In accordance, Ananda Lages Rodrigues et al. reported that neonatal overfeeding reduced NPY hypothalamic content [44]. Additionally, in females, it is noted increased levels of glutamatergic synapse, namely vGLUT1 and PSD95 levels. Indeed, postnatal overnutrition has been shown to affect neuronal projections and synaptic transmission in females, due to an increase in excitatory transmission and lower leptin levels [45].

NPY is also involved in the regulation of emotional, affective behaviour and stress responses [22]. Indeed, NPY has similarities with GABAergic agents such as benzodiazepines in reducing CNS activity, and low levels of NPY have been associated with psychiatric disorders including anxiety and depression [46]. NPY controls anxiety primarily via the activation of postsynaptic Y1R and Y5R and of presynaptic Y2R, inducing anxiolytic and anxiogenic effects, respectively [21,22,47]. NPY1R is regulated by the early maternal environment and is linked to anxiety and stress reaction regulation, being related to appetite stimulation [22,48]. Our results showed that, in the prefrontal cortex, male offspring presented lower NPY1R and NPY2R, accompanied by lower GHS-R1α levels, whereas females only presented lower NPY1R levels, suggesting that postnatal overfeeding modulates NPY signalling in this region. To our knowledge, this is the first study to explore the role of NPY signalling in the prefrontal cortex in postnatal overweight animals. Accordingly, in a model of maternal deprivation, it has been shown that offspring were less anxious and NPY1R is highly expressed in the prefrontal cortex. Its manipulation elicit GABAergic activity, suggesting that NPY1R could modulate inhibitory synapses [13]. Indeed, our results seem to be in concordance with that. Although it is noted higher GABA_A_ receptor levels, its scaffold protein gephyrin presented lower levels, suggesting that its inhibitory activity may be reduced, and the increased receptor expression may be a compensatory mechanism [49]. Furthermore, in overweight offspring due to a high-fat diet consumption, reduced prefrontal cortex GABA levels were noted together with deficits in memory and learning processes, suggesting that deficits in GABAergic neurotransmission may be associated with working and spatial memory impairments throughout life and delayed neurodevelopment achievements [14].

In the hippocampus, only females presented higher NPY2R levels accompanied by lower NPY1R and GHS-R1α levels, together with lower GABAergic proteins—vGAT and GABA_A_—suggesting that NPY1R also mediates GABAergic action in this region. Furthermore, upon overnutrition conditions, females are deeply affected in both the hippocampus and prefrontal cortex regions, while males are more susceptible to prefrontal cortex changes. Interestingly, maternal high-fat diet consumption seems to decrease NPY1R levels together with behavioural impairments and volumetric changes, highlighting the heightened effect of NPY on hippocampal learning–memory and neurogenesis-related receptors [24]. Such region-specificities were also observed for ghrelin. Our results showed that neonatal overfeeding induced lower ghrelin receptor levels in the male prefrontal cortex, whereas females had a reduction in the hippocampus. Ghrelin has been described as an important modulator of neuronal activity and synaptic plasticity, especially in glutamate receptors involved in excitatory synapses, having an important role in memory function [18]. Few studies have been addressed the role of ghrelin in these brain regions, showing that caloric restriction or ghrelin injections resulted in anxiolytic and anti-depressant behaviour in non-stressed mice [50,51]. Given the NPY regulation by ghrelin, we may hypothesize that at least part of such effects may be related to NPY upregulation. Accordingly, neonatal overfeeding may trigger changes in central ghrelin levels, contributing to the onset of stress-related disorders. Some studies have addressed that ghrelin has an important role in memory and learning retention, although our results did not show alterations in learning and memory tasks or major changes in ghrelin.

## 5. Conclusions

Our work provides new insights into the sex-specific impact of neonatal overfeeding on neurodevelopment and behaviour (Figure 9). We demonstrate that female offspring are more susceptible to neurodevelopmental delays, although, during the juvenile period, both males and females developed an anxious-like phenotype. Furthermore, we point out the role of NPY as a potential bridge between energy balance mechanisms and emotional behaviour, demonstrating that, in overfeeding conditions, both males and females have brain-region-specific alterations in the NPY/ GHS-R1α system, which contribute to metabolic and behavioural disruptions later in life. Furthermore, we hypothesise that the NPY system, particularly NPY1R, modulates GABA_A_ receptor levels, contributing to the disruption of neurotransmitter balance, with long-term consequences on offspring behaviour. Notably, we demonstrate that disturbances during the postnatal period imprint metabolic and neurobehavioural alterations in offspring, which may increase the risk of metabolic syndrome and psychiatric disorders later in life.

## Figures and Tables

**Figure 1 nutrients-15-03581-f001:**
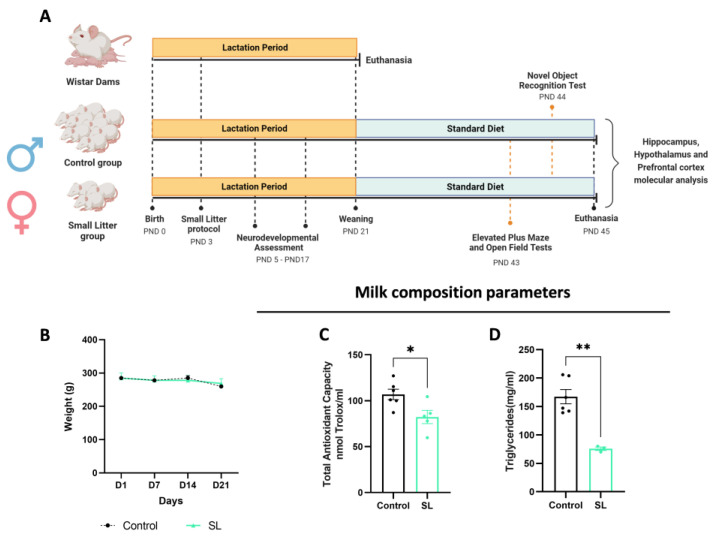
Postnatal overfeeding alters milk composition and quality. Postnatal overfeeding experimental design (**A**). Maternal body weight gain curve during the breastfeeding period (**B**). Lower total antioxidant capacity (**C**) and triglycerides (**D**) levels were noted in breastmilk content of SL dams at weaning. The results are shown as mean ± SEM of 3 to 5 dams per group and unpaired *t*-test was conducted to compare the experimental groups.*, *p* < 0.05; **, *p* < 0.01.

**Figure 2 nutrients-15-03581-f002:**
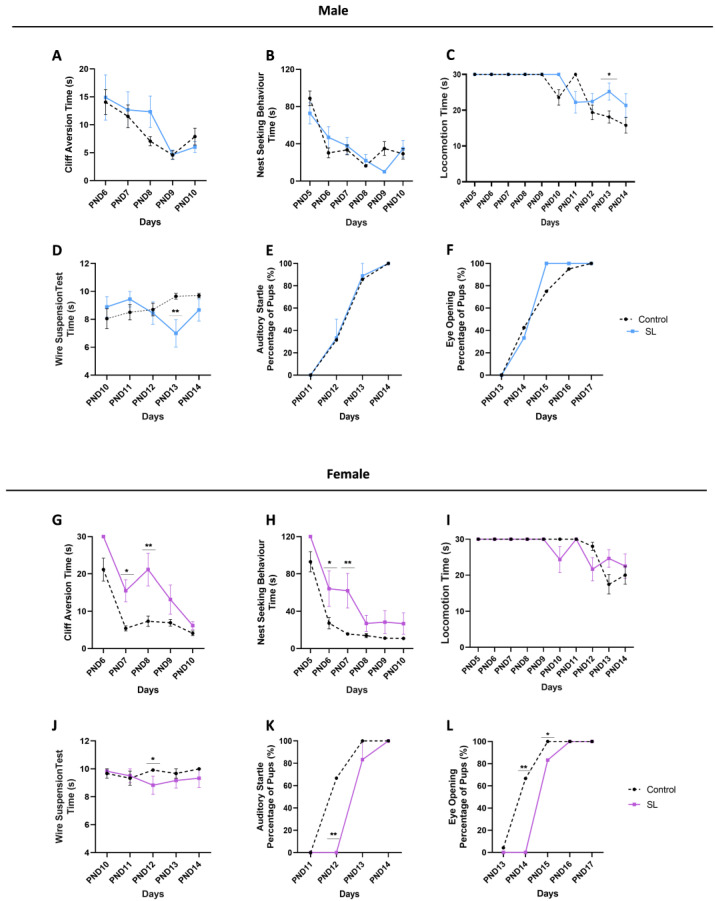
Postnatal overfeeding increases female susceptibility to neurodevelopmental delay. Pups were tested in several behavioural tests from PND5 until PND17. Male offspring showed no alterations in tests dependent on vestibular system (**A**) and olfactory maturation (**B**). A transitory impairment in locomotion performance (**C**) and upper limbs strength (**D**) was detected at PND13. Additionally, no alterations were observed in auditory ability (**E**) and eye-opening day (**F**). Regarding females, impairments in vestibular system maturation (**G**) and olfactory discrimination (**H**) were detected without alterations in locomotor performance (**I**). Furthermore, females presented less strength in their upper limbs (**J**) and a significant delay in their auditory ability (**K**) and eye-opening day (**L**). The results are shown as mean ± SEM of 9 to 20 males per group and 6 to 12 females per group. Unpaired *t*-test was conducted to compare the experimental groups. *, *p* < 0.05; **, *p* < 0.01.

**Figure 3 nutrients-15-03581-f003:**
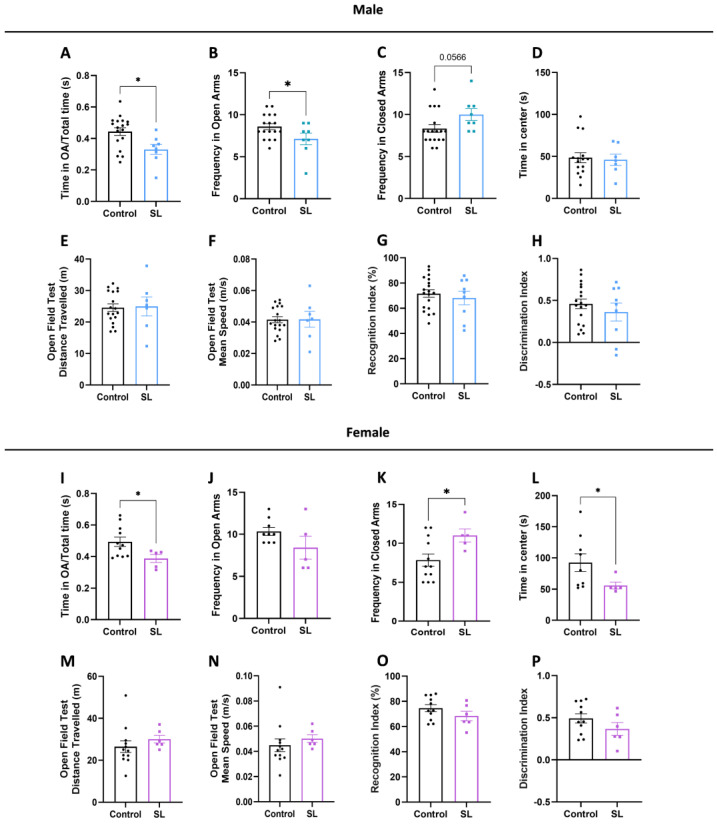
Postnatal overfeeding induces anxious-like behaviour in both males and females. At adolescence (PND43), offspring were tested in EPM test to evaluate anxious-like behaviour. Males remained less time exploring the open arms in the EPM arena (**A**), entering less frequently in the open arms (**B**), and spent more time in the closed arms (**C**). At PND43, male animals performed the OPF test to evaluate explorative behaviour and locomotion, and no alterations were noted regarding the time of exploring the center of the arena (**D**), as well as in the distance travelled (**E**) and mean speed (**F**). At PND44, the offspring was tested in NOR test to evaluate short-memory retention, although no alterations were observed in the RI (**G**) and DI (**H**). Similarly, to males, females spent less time exploring the open arms in the EPM arena (**I**), without alterations in the number of entries in the open arms (**J**), although a higher frequency of entry in the closed arms was observed (**K**). Furthermore, female offspring spent less time exploring the center of the OPF arena (**L**), without alterations in the distance travelled (**M**) and mean speed (**N**). Additionally, no alterations were noted in RI (**O**) and DI (**P**) regarding NOR test analysis. The results are shown as mean ± SEM of 9 to 20 males per group and 6 to 12 females per group. Unpaired *t*-test was conducted to compare among the experimental groups. *, *p* < 0.05.

**Figure 4 nutrients-15-03581-f004:**
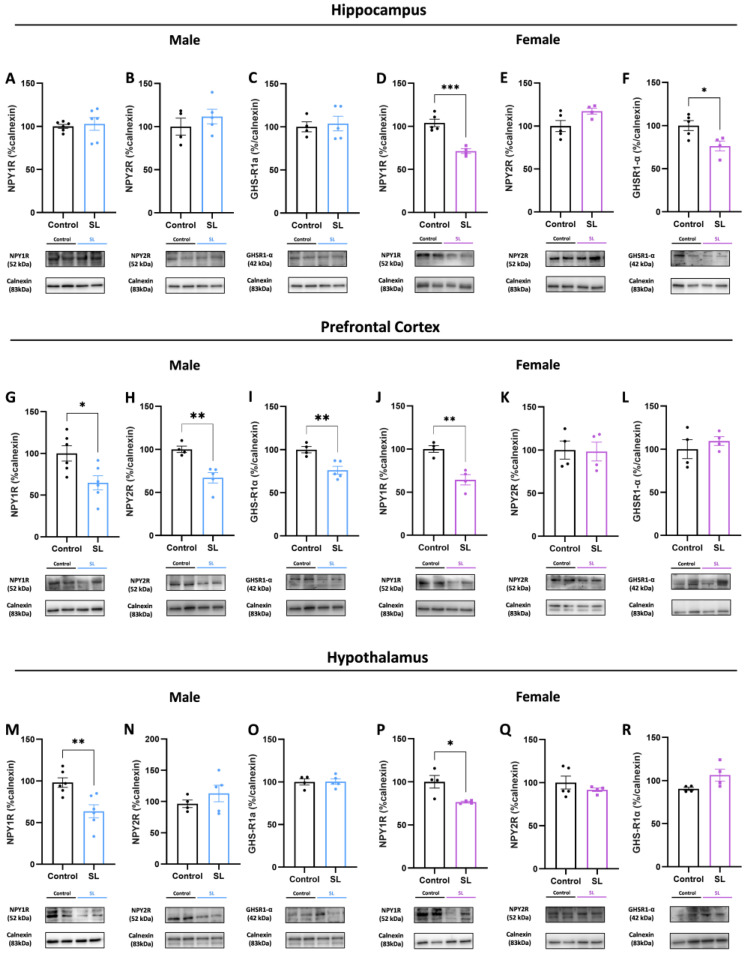
Postnatal overfeeding downregulates NPY signalling at central level according to sex and brain region specificities. No alterations were noted in NPY1R (**A**), NPY2R (**B**), and GHS-R1α (**C**) protein levels in males hippocampus, while, in females, lower levels of NPY1R (**D**) and GHS-R1α were noted (**F**), without alterations in NPY2R levels (**E**). In the male prefrontal cortex, lower NPY1R (**G**), NPY2R (**H**), and GHS-R1α (**I**) were observed. On the other hand, females presented lower NPY1R levels (**J**), without significant alterations in NPY2R (**K**) and GHS-R1α (**L**) levels. In the hypothalamus, lower NPY1R was noted in both males (**M**) and females (**P**), without alterations in NPY2R (**N**,**Q**) and GHS-R1α levels (**O**,**R**). The results are shown as mean ± SEM of 4 to 6 animals per group for Western blot analysis. For statistical analysis, an unpaired *t*-test was conducted to compare the experimental groups. *, *p* < 0.05; **, *p* < 0.01; ***, *p* < 0.001.

**Figure 5 nutrients-15-03581-f005:**
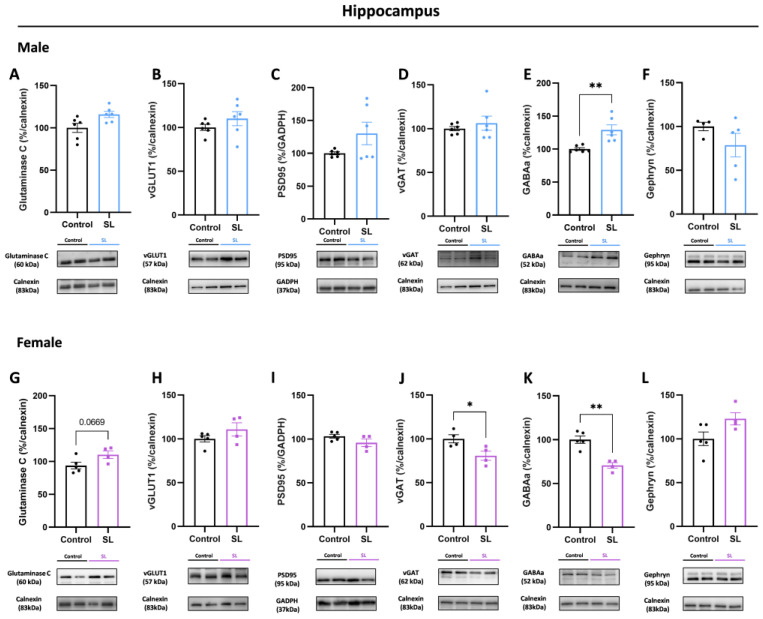
Postnatal overfeeding induces hippocampal GABA_A_ receptor alterations in a sex-dependent manner. Male offspring from SL group presented no significant alterations in glutaminase C (**A**), vGLUT1 (**B**), and PSD95 (**C**) levels. Regarding GABAergic synapse, postnatal overfeeding induced higher GABA_A_ receptor levels (**E**), although no alterations were observed in vGAT (**D**)at the presynaptic site and in gephyrin (**F**) at the postsynaptic site. Similarly, to males, females presented a trend to higher glutaminase C (**G**), without alterations in vGLUT1 (**H**) and PSD95 (**I**) proteins levels. In the inhibitory synapse, lower vGAT (**J**) and GABA_A_ (**K**) receptor levels were observed without alterations in gephyrin (**L**) levels. The results are shown as mean ± SEM of 4 to 6 animals per group for Western blot analysis. For statistical analysis, unpaired *t*-test was conducted to compare the experimental groups. *, *p* < 0.05; 2 symbols, *p* < 0.01.

**Figure 6 nutrients-15-03581-f006:**
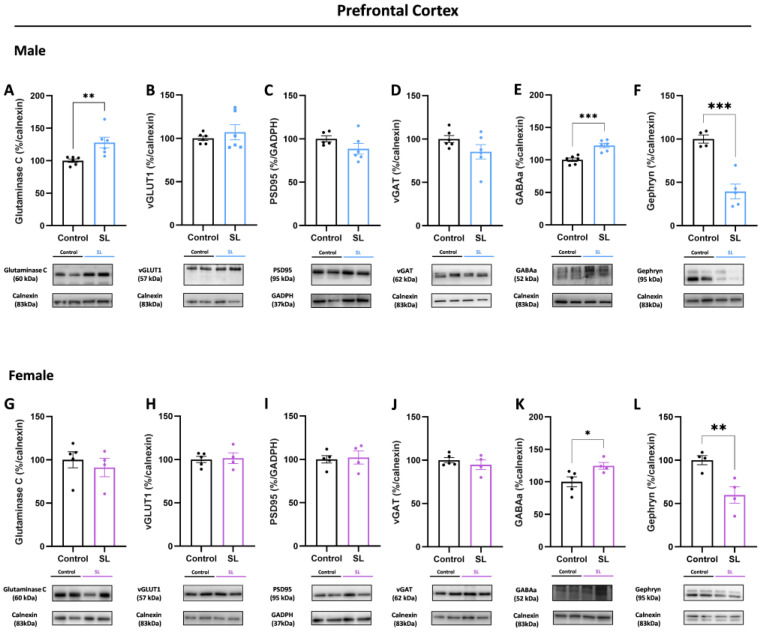
Male and female offspring presented postsynaptic GABAergic changes at the prefrontal cortex upon postnatal overfeeding conditions. In the prefrontal cortex, males from SL group had significantly higher glutaminase C (**A**) levels without alterations in vGLUT1 (**B**) and PSD95 (**C**) protein levels. Although no major differences were noted at the presynaptic level, in vGAT (**D**) protein levels, it was observed higher GABA_A_ receptor (**E**) levels, together with lower gephyrin (**F**) levels at the postsynaptic site. Regarding females, no alterations were detected in glutaminase C (**G**), vGLUT1 (**H**), and PSD95 (**I**) proteins. At the GABAergic synapse, no alterations were observed in vGAT (**J**) protein levels, however, higher levels of GABA_A_ (**K**) receptor were noted together with lower gephyrin (**L**) levels. The results are shown as mean ± SEM of 4 to 6 animals per group for Western blot analysis. For statistical analysis, unpaired *t*-test was conducted to compare the experimental groups. *, *p* < 0.05; **, *p* < 0.01; ***, *p* < 0.001.

**Figure 7 nutrients-15-03581-f007:**
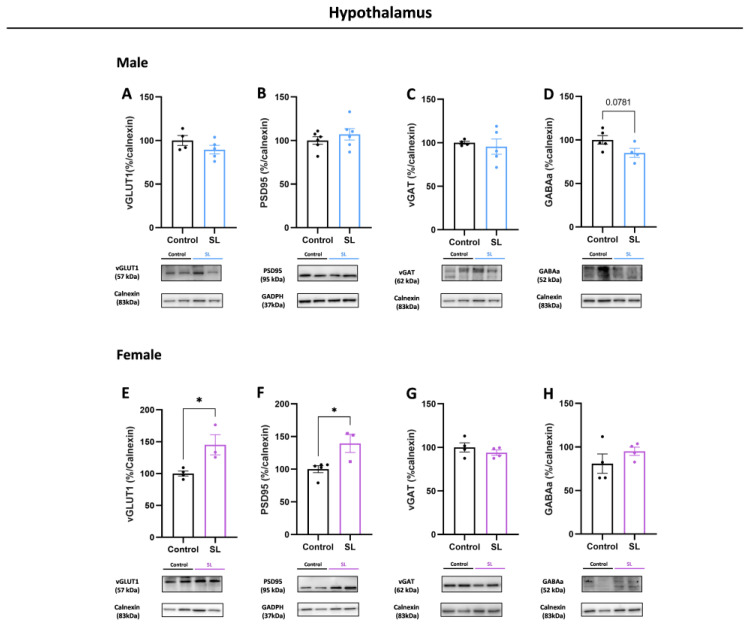
Postnatal overfeeding changes glutamatergic synapse levels in females but not in males. Males from SL group had no major differences in vGLUT1 (**A**), PSD95 (**B**), vGAT (**C**), and GABA_A_ receptor (**D**) protein levels. In females, at the excitatory synapse, higher levels of vGLUT1 (**E**) and PSD95 (**F**) were observed, without alterations in vGAT (**G**) and GABA_A_ (**H**) receptor levels. The results are shown as mean ± SEM of 3 to 5 animals per group for Western blot analysis. For statistical analysis, an unpaired *t*-test was conducted to compare the experimental groups. *, *p* < 0.05.

**Figure 8 nutrients-15-03581-f008:**
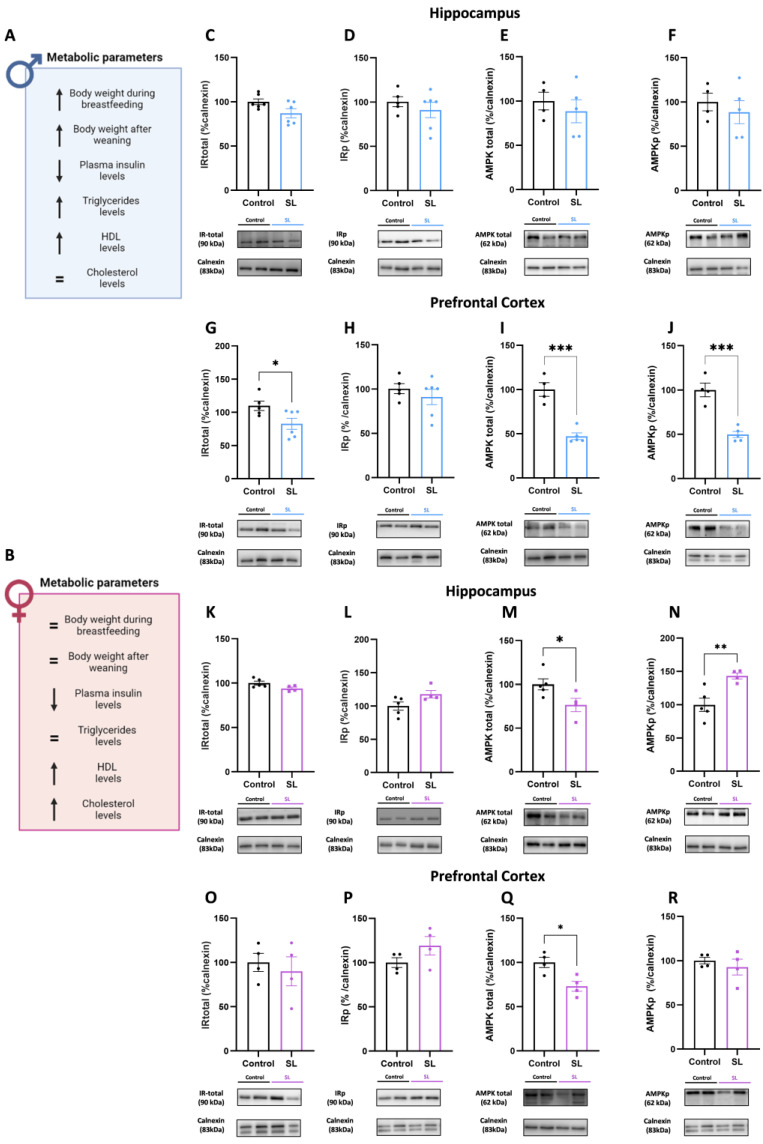
Postnatal overfeeding alters AMPK activation in a sex and brain-region-specific manner. Metabolic parameters of males (**A**) and females (**B**) under postnatal overfeeding conditions. In male hippocampus, no alterations were observed in proteins involved in insulin signalling pathway, namely total IR (**C**), IRp (**D**), total AMPK (**E**), and phosphorylated AMPK (**F**). In the prefrontal cortex, lower levels of total IR (**G**), total AMPK (**I**), and its phosphorylation form (**J**) were noted, without alterations in phosphorylated IR (**H**). In the female hippocampus, no alterations were observed in total IR (**K**) levels and its phosphorylated form (**L**), although lower total AMPK (**M**) levels were noted, accompanied by higher levels of its phosphorylation (**N**). In the female prefrontal cortex, no alterations were observed in total (**O**) and phosphorylated (**P**) IR levels; although lower total (**Q**) AMPK was noted, no changes were observed regarding AMPK activation (**R**). The results are shown as mean ± SEM of 4 to 6 animals per group for Western blot analysis. For statistical analysis, unpaired *t*-test was conducted to compare the experimental groups. *, *p* < 0.05; **, *p* < 0.01; ***, *p* < 0.001.

**Figure 9 nutrients-15-03581-f009:**
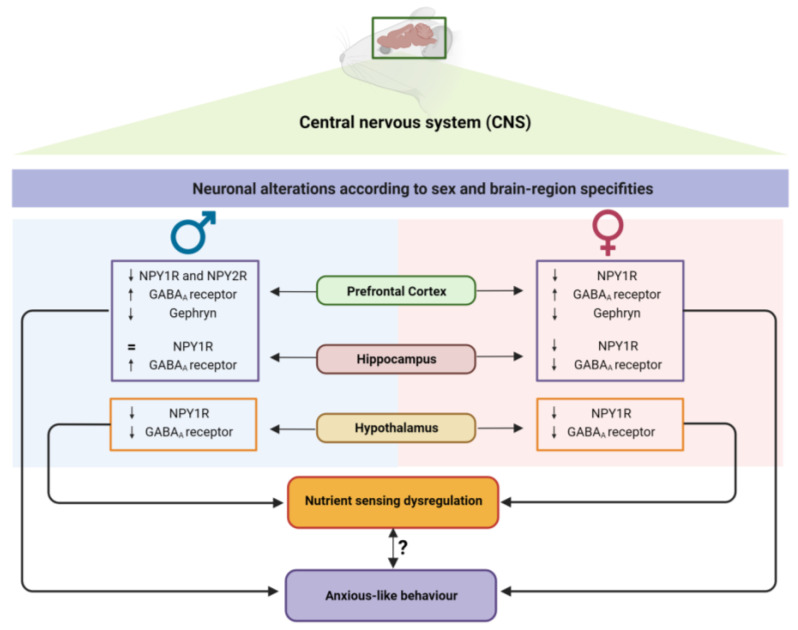
Postnatal overfeeding modulates offspring behaviour and nutrient-sensing mechanisms in a sex- and brain-region-specific manner due to NPY/GABA dysregulation.

**Table 1 nutrients-15-03581-t001:** Primary and secondary antibodies used in Western blot.

Primary Antibody	Molecular Weight	Secondary Antibody	Company
Anti-IR-total	~90 kDa	Anti-Rabbit	Cell Signaling
Anti-IR (phospho Y1361)	~90 kDA	Anti-Rabbit	Abcam
Anti-AMPK-total	~60 kDa	Anti-Rabbit	Cell Signaling
Anti-AMPK (phospho Y172)	~60 kDa	Anti-Rabbit	Cell Signaling
Anti-Glutaminase C	~66 kDa	Anti-Rabbit	Cell Signaling
Anti-vGLUT1	~57 kDa	Anti-Rabbit	Abcam
Anti-vGAT	~62 kDa	Anti-Mouse	Abcam
Anti-GABA_A_	~52 kDa	Anti-Rabbit	Abcam
Anti-PSD95	~95 kDa	Anti-Rabbit	Cell Signaling
Anti-NPY1R	~52 kDa	Anti-Sheep	Bio-Rad
Anti-NPY2R	~52 kDa	Anti-Goat	Abcam
Anti-GHSR1α	~42 kDa	Anti-Rabbit	Abcam
Anti-Gephyrin	~95 kDa	Anti-Mouse	ThermoFisher
Anti-Calnexin	~83 kDa	Anti-Goat	Sicgen
Anti-GADPH	~37 kDa	Anti-Goat	Sicgen

## Data Availability

The datasets generated during and/or analysed during the current study are available from the corresponding author upon reasonable request.

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
