# Peer review of "Postnatal Overfeeding in Rodents Induces a Neurodevelopment Delay and Anxious-like Behaviour Accompanied by Sex- and Brain-Region-Specific Synaptic and Metabolic Changes"

_nutrients, 2023, doi:10.3390/nu15163581_

Round 1

Reviewer 1 Report

Using Wistar rats as a model, this paper simulates the effects of neonatal overfeeding on neurodevelopment and behavior and the sex specificity of these effects. The results showed that both female and male offspring who were overfed developed anxiety-like phenotypes at an early age, but female offspring were more susceptible to neurodevelopmental delays. This study shows that the level of nutrition fed after birth is not only related to the metabolism at the time but also has a certain impact on the future development of the offspring.

Concerns: some WB figures in the results of the article do not reflect the trend well. For example, the SL group of Figure is not significantly down-regulated and the trend of the Control group is not uniform, the SL group of Figure4G is not significantly down-regulated and the trend of SL group is not uniform, and the intra-group trend of the two groups of Figure4N is not uniform. In my opinion, it is impossible to compare the changes between the two groups. The WB bands in Figure5A cannot show the changes between the two groups; the trends within the two groups in Figure6F are not uniform; the trends between the two groups in Figure7E are not uniform; the downregulation of the SL group in Figure8Q is not obvious and the trends in the SL group are not uniform. In addition, the bands of Figure4R, Figure6E and Figure6K are very fuzzy, and the trend cannot be judged clearly. Finally, many WB charts in this paper have many mixed bands, and it is impossible to judge which one is the target band.

The experimental design of the whole paper is relatively complete, but the credibility of the experimental results involving WB is not high. It is suggested to modify the above problems, and some experiments can be redone to show your results with more convincing result graphs.

Author Response

We are grateful to all the reviewers for the attentive reading of our work and their constructive and suggestive comments. Below is a point-by-point response to the reviewers’ comments. We updated the manuscript according to the reviewers’ comments and have included additional information (in track changes). We believe the manuscript is now improved and hope that it meets your expectations.

Reviewer 1: Using Wistar rats as a model, this paper simulates the effects of neonatal overfeeding on neurodevelopment and behavior and the sex specificity of these effects. The results showed that both female and male offspring who were overfed developed anxiety-like phenotypes at an early age, but female offspring were more susceptible to neurodevelopmental delays. This study shows that the level of nutrition fed after birth is not only related to the metabolism at the time but also has a certain impact on the future development of the offspring.

Concerns: some WB figures in the results of the article do not reflect the trend well. For example, the SL group of Figure is not significantly down-regulated and the trend of the Control group is not uniform, the SL group of Figure4G is not significantly down-regulated and the trend of SL group is not uniform, and the intra-group trend of the two groups of Figure4N is not uniform. In my opinion, it is impossible to compare the changes between the two groups. The WB bands in Figure5A cannot show the changes between the two groups; the trends within the two groups in Figure6F are not uniform; the trends between the two groups in Figure7E are not uniform; the downregulation of the SL group in Figure8Q is not obvious and the trends in the SL group are not uniform. In addition, the bands of Figure4R, Figure6E and Figure6K are very fuzzy, and the trend cannot be judged clearly. Finally, many WB charts in this paper have many mixed bands, and it is impossible to judge which one is the target band.

Thank you for your suggestions. We have carefully reanalysed all the quantifications and western blot images to improve their quality and validation. We also detected that we send the file with Western Blot results without the representative molecular weight standard of each membrane, so we now uploaded a new file showing that and highlighting (red bar) the specific bands quantified. Regarding figure 4G and 4N we reanalysed all the quantifications and corrected the background of all representative membranes. In figure 4R we ameliorate the signal, although the control bands for calnexin (protein of control) show similarity, there is variability in the SL group, demonstrating no overall alterations in ghrelin receptor levels.  In figures 5A, 6E, and 6F we also quantified each membrane, and it was selected another representative WB band. Finally, in figures 7E and 8Q the background was also corrected.

The experimental design of the whole paper is relatively complete, but the credibility of the experimental results involving WB is not high. It is suggested to modify the above problems, and some experiments can be redone to show your results with more convincing result graphs.

Reviewer 2 Report

The authors are concerned with neurodevelopmental changes resulting from nutritional disturbances following birth and any correlation with long-lasting effects on behavioral changes into adulthood. Investigation of this correlation stems from the direct connection between neuroendocrine pathways and nutrient-sensing balance. Neuroendocrine pathways play an essential role in neuronal development and behavioral alterations. The authors wished to evaluate these changes in diet between the two sexes and investigate the underlying mechanisms within the central nervous system (CNS). The authors investigated in mice postnatal day 3 (PND3) with an N=3. Neurodevelopment tests were performed in infancy, and behavioral tests like locomotion, anxiety, and memory were performed in adolescence.

Furthermore, the brain was removed during adolescence for western blot analysis of the hippocampus, hypothalamus, and prefrontal cortex. Female test groups exhibited impaired auditory response, eye-opening, olfactory discrimination, and vestibular system development in infancy. In male offspring, the test group exhibited a delay in locomotor performance. Both offspring had reduced upper body strength. At adolescence, both test groups exhibited anxious-like behavior. Analysis of the resected brain indicated females had lower GABAa in the hippocampus and higher levels of vGLUT1 and PSD95 in the hypothalamus. The males exhibited elevated levels of GABAa. Both groups had higher GABAa receptor levels in the prefrontal cortex and lower NPY1R levels across all regions.  

One strength of the paper was the number of behavioral tests and validation of the tests. Many potential caveats in each condition were also addressed by the authors and corrected to ensure accurate behavioral measurement. The authors were also very thorough in their western blot protocols and ensured the antibodies' accuracy for western blot analysis of the samples. Another strength is that the statistical analysis exhibited significance with minimal variation between their measurements and was significant with a simple unpaired t-test. Furthermore, the authors do an excellent job summarizing the data in Figures 8 and 9.

One weakness is in the western blot Figure 4R. The control representative blot seemed not to transfer correctly, and when compared to the results of the bar graph gives the impression that some of the points might be off. Furthermore, the quantification for some of the blots was unclear. When multiple bands are present, the authors should identify the bands they specifically quantified or whether rit was the sum of all bands in Figures 4N and 6F. Furthermore, some bands seem completely cropped off, like in 6K and 7H.

The paper is an exciting investigation into the connection between overfeeding and the role of neuronal development and long-term behavioral changes. While the authors conducted excellent behavior studies, the represented western blots in the figures could be improved upon or warrant further analysis. Overall, I think the study is interesting and parallelles previous nutritional behavioural studies while supporting a biochemical rationale for these changes. I believe this paper should be accepted with some minor revisions.

There were no major issues in the language but the authors should take some time to go through and address articles and prepositions in some cases.

Author Response

The authors are concerned with neurodevelopmental changes resulting from nutritional disturbances following birth and any correlation with long-lasting effects on behavioral changes into adulthood. Investigation of this correlation stems from the direct connection between neuroendocrine pathways and nutrient-sensing balance. Neuroendocrine pathways play an essential role in neuronal development and behavioral alterations. The authors wished to evaluate these changes in diet between the two sexes and investigate the underlying mechanisms within the central nervous system (CNS). The authors investigated in mice postnatal day 3 (PND3) with an N=3. Neurodevelopment tests were performed in infancy, and behavioral tests like locomotion, anxiety, and memory were performed in adolescence. Furthermore, the brain was removed during adolescence for western blot analysis of the hippocampus, hypothalamus, and prefrontal cortex. Female test groups exhibited impaired auditory response, eye-opening, olfactory discrimination, and vestibular system development in infancy. In male offspring, the test group exhibited a delay in locomotor performance. Both offspring had reduced upper body strength. At adolescence, both test groups exhibited anxious-like behavior. Analysis of the resected brain indicated females had lower GABAa in the hippocampus and higher levels of vGLUT1 and PSD95 in the hypothalamus. The males exhibited elevated levels of GABAa. Both groups had higher GABAa receptor levels in the prefrontal cortex and lower NPY1R levels across all regions.  

One strength of the paper was the number of behavioral tests and validation of the tests. Many potential caveats in each condition were also addressed by the authors and corrected to ensure accurate behavioral measurement. The authors were also very thorough in their western blot protocols and ensured the antibodies' accuracy for western blot analysis of the samples. Another strength is that the statistical analysis exhibited significance with minimal variation between their measurements and was significant with a simple unpaired t-test. Furthermore, the authors do an excellent job summarizing the data in Figures 8 and 9.

One weakness is in the western blot Figure 4R. The control representative blot seemed not to transfer correctly, and when compared to the results of the bar graph gives the impression that some of the points might be off. Furthermore, the quantification for some of the blots was unclear. When multiple bands are present, the authors should identify the bands they specifically quantified or whether rit was the sum of all bands in Figures 4N and 6F. Furthermore, some bands seem completely cropped off, like in 6K and 7H.

Thank you for your suggestions. Indeed, the western blot of figure 4R (ghrelin receptor in female offspring hypothalamus) shows some variability between the groups, however, we repeated this western blot, and the result was consistent, showing no significant alterations in this region, which allowed us to hypothesize that NPY1R alterations are not caused by ghrelin alterations. Regarding the quantifications, we carefully reanalysed all the quantification and western blot images. One particularity of NPY and ghrelin receptors is the appearance of multiple bands in western blot membranes. All the data shown refer to the quantification of the bands in the expected molecular weight and no other bands from unknown origin (section 2.5 of materials and methods in table 1). We detected that we sent the file with Western Blot results without the representative molecular weight of each membrane, so we now uploaded a new file showing that and highlighting (red bar) the specific bands quantified. Regarding figures 6K and 7H, we selected different representative membranes.

The paper is an exciting investigation into the connection between overfeeding and the role of neuronal development and long-term behavioral changes. While the authors conducted excellent behavior studies, the represented western blots in the figures could be improved upon or warrant further analysis. Overall, I think the study is interesting and parallelles previous nutritional behavioural studies while supporting a biochemical rationale for these changes. I believe this paper should be accepted with some minor revisions.

Reviewer 3 Report

This manuscript presents an interesting concept on the role of over-feeding and neurodevelopment and behavior. The model used to presumably cause over-feeding involves culling the litters severely, which resulted in substantial biochemical changes in maternal milk, most notably reductions in anti-oxidants. The strength of the work rests in the neurobehavioral studies. The weaknesses pertain to most other components of the work.  1) Asking the readers to rely on prior publications as proof that the model indeed caused overfeeding is not acceptable. The body weight changes over time for males and females in each group must be shown. We don't actually know that the "overfed" pups gained more weight or even if they had been overfed or if milk production declined with litter size; 2) Another concern is the definition of adolescence. It is unclear what age the animals were studied for molecular assays of their brains. Adolescence is not defined. Below P30 is NOT adolescent. Transparency is needed. All figure legends should indicate the ages of the offspring. 3) The authors ignore the fact that all parameters (behavioral) self-correct over time, suggesting the absence of long-term effects, and, instead, just transient impacts. 4) The authors do not consider that the alterations in maternal milk could be the driver of brain functional and molecular abnormalities. There are no controls for anti-oxidant reduction and lipid changes that could impact brain development; 5) Maternal behavior was not assessed. How the dams interacted with young pups could influence behavior and development. Severe culling of the litters could impact how the dams treat their offspring; 6) There are obvious problems with the quality of the Westerns--smudginess, uneven loading, background; 7) it does not appear that the original Westerns were each re-probed for control signals since the gel artifacts differed indicating that the blots had not been re-probed--specific signals showed lane tilts, frowns,  and other distortions not present in the control bands; 8) instead of normalizing everything to 100%, it would be more transparent to show actual normalized densitometry signals as they could also be used to compare levels of different protein expression within the same regions; 9) Statistical comparisons require corrections for false discovery rates due to repeated analysis of the same samples; 10) the schemes and conclusions are too speculative given the data provided.

Author Response

This manuscript presents an interesting concept on the role of over-feeding and neurodevelopment and behavior. The model used to presumably cause over-feeding involves culling the litters severely, which resulted in substantial biochemical changes in maternal milk, most notably reductions in anti-oxidants. The strength of the work rests in the neurobehavioral studies. The weaknesses pertain to most other components of the work.  1) Asking the readers to rely on prior publications as proof that the model indeed caused overfeeding is not acceptable. The body weight changes over time for males and females in each group must be shown. We don't actually know that the "overfed" pups gained more weight or even if they had been overfed or if milk production declined with litter size;

Thank you for your commentary and critical suggestions. We have recently published an original paper describing the effects of the small litter procedure, in the animals included in this present study, considering their metabolic profile and body weight composition [1]. Since the animals were the same and the article is already published, we herein only refer to the published work. We previously demonstrated that male offspring have increased body weight gain during the breastfeeding period that is maintained after weaning, accompanied by lower plasma insulin levels, and increased triglycerides and HDL levels. Regarding females, although no alterations were observed in body weight composition, lower plasma insulin levels and increased HDL and cholesterol levels were detected. Therefore, we don’t show these data in the present manuscript because as we mentioned it has been already published and we considered that it is not scientifically ethical to include published data herein, which is a violation of journal’s policy. Therefore, we have summarized these results in figures 8A and 8B showing that the SL procedure induces overweight in a sex-dependent manner. During the experiment, as described in section 2.1 of material and methods, all the pups from the control and SL groups were maintained with the same conditions after weaning (ad libitium access to standard diet and water), therefore the weight gain is due to an increase in milk uptake and not to a different after-weaning manipulation. It has also been demonstrated that a small litter exhibits a higher milk intake per pup (around 1.5-fold) than a normal-sized litter [2]. Remarkably, it is important to highlight that we have in consideration that the lower number of pups could affect and reduce the production of milk. Therefore, the litter size reduction was performed at PND3 to avoid that.

2) Another concern is the definition of adolescence. It is unclear what age the animals were studied for molecular assays of their brains. Adolescence is not defined. Below P30 is NOT adolescent. Transparency is needed. All figure legends should indicate the ages of the offspring.

Thank you for your suggestion. According to the literature, by definition, in rodent models adolescence period ranged from weaning (PND21) to adulthood (PND60). In a more detailed division, this period can be divided into three stages, namely early adolescence or juvenile (PND21-PND34), middle adolescence (PND34-PND46), and late adolescence or young adulthood (PND46-PN59) [3]. As we described in section 2.3 of material and methods, offspring were tested in a battery of neurodevelopmental tests during infancy, namely at PND5 to PND17, which results are detailed showed in figure 2.  Between PND43 and PND44, offspring were also tested in several behavioral tests related to anxiety, memory, and explorative behaviour, which is considered the middle adolescence period, as described in section 2.4 of the materials and methods sections. At PND45 the offspring were euthanized, and the different brain regions were collected for molecular analysis as detailed in sections 2.1 and 2.5 of materials and methods and schematized in figure 1. 

3) The authors ignore the fact that all parameters (behavioral) self-correct over time, suggesting the absence of long-term effects, and, instead, just transient impacts.

It is true that some behavioral parameters can be self-corrected over time. However, it is not necessarily accurate to suggest the absence of long-term effects and only transient impacts.  The self-correction of behavioural parameters can depend on the behaviour, age, and the context in which it occurs. For instance, adverse childhood events, such as exposure to chronic stress or early-life traumatic experiences can lead to persistent behavioral patterns, and changes in brain structure with profound psychological long-term consequences. The aim of this work was to study the impact of the small litter procedure on offspring neurodevelopment and behaviour, which comprises infancy and adolescence period, respectively. Therefore, we did not focus on studying long-term effects but, given the results obtained, it would be interesting to study them in the future.

4) The authors do not consider that the alterations in maternal milk could be the driver of brain functional and molecular abnormalities. There are no controls for anti-oxidant reduction and lipid changes that could impact brain development;

Thank you for your suggestion. We evaluated some parameters related to milk composition, namely total antioxidant capacity and triglycerides levels, both of which were reduced, suggesting that small litter procedure drives alterations in maternal milk parameters. The aim of this work was to evaluate the behavioural and neurodevelopment consequences of excessive nutritional intake in the early life, using the small litter as a model. The putative impact on mechanisms related to energy balance and neurotransmitters balance were investigated, but we didn’t explore potential molecular alterations regarding oxidative stress. We analysed glyoxalase 1 and catalase levels and no significant alterations were observed, as shown in the figure below.

5) Maternal behavior was not assessed. How the dams interacted with young pups could influence behavior and development. Severe culling of the litters could impact how the dams treat their offspring;

Thank you for your suggestive commentary. This is an excellent point; the maternal bond is crucial for shaping and modulating offspring behaviour during life. Some studies have demonstrated that litter size reduction can cause important maternal alterations like increased maternal care, longer breastfeeding and reduced total time out of the nest [4]. This is supported by the fact that the severe culling of the litter makes the dams more protective and caring than those who have a normal litter. Indeed, we don’t assess maternal behaviour since we were focused in evaluating offspring neurodevelopment, and the manipulation/separation of the dams from their pups could cause profound changes in their development. However, we have discussed this hypothesis (line 526-532) and the role of maternal behaviour on offspring behaviour and neurodevelopment during life.

6) There are obvious problems with the quality of the Westerns--smudginess, uneven loading, background; 7) it does not appear that the original Westerns were each re-probed for control signals since the gel artifacts differed indicating that the blots had not been re-probed--specific signals showed lane tilts, frowns, and other distortions not present in the control bands;

Thank you for your commentary. Each original membrane containing the protein of interest was then re-probed for loading control protein. The bands considered to quantify the protein of interest, or the control loading (calnexin or GADPH) are present at different molecular weight (as described in Table 1 of the materials and methods section), so it is normal that some imperfections are not visible. We have carefully reanalysed all the quantifications and western blot images to improve their quality and validation. We also detected that we send the file with Western Blot results without the representative molecular weight standard of each membrane, so we now uploaded a new file showing that and highlighting (red bar) the specific bands quantified.

8) instead of normalizing everything to 100%, it would be more transparent to show actual normalized densitometry signals as they could also be used to compare levels of different protein expression within the same regions;

Thank you for your suggestion. Western blot analysis is a semi-quantitative technique calculated through the normalization of the interested proteins to the loading control (calnexin or GADPH). This data analysis avoids any discrepancies that may occur during the experiments. Since this analysis is based on the ratio, the normalization to percentage of control is more correct, and avoid the bias originated by different exposures degrees and signal intensity between membranes. Nevertheless, the calculation for each protein of interest is normalized for the control group (n=6 in males offspring, since it was used two membranes (3 controls in each membrane) and n=5 in females offspring). In this way, the calculation was based on the average of the controls group to normalize and validate the quantification. For that reason, in some proteins, there is variability in the control group (Figure 6G, for example) and the range of variation can be different.

9) Statistical comparisons require corrections for false discovery rates due to repeated analysis of the same samples;

Thank you for your suggestion, but no sample repetition was made. Each lane represents a different animal and, consequently, a different sample. Therefore, in each protein analysed, we considered 6 different male offspring animals per group (control and SL groups), 5 female offspring in the control group, and 4 female offspring from SL group. However, in the hypothalamus, the figures 7E and 7F have a lower number of animals (3 per group) since it is a region with a low sample volume and a limited number of tissues was available for each experiment. In all figure legends, we have described the number of animals used for each experiment, and we have now clarified this information in section 2.6 of materials and methods.

10) the schemes and conclusions are too speculative given the data provided.

Thanks for your comment. Figures 8A and 8B are schematic figures with the characterization of the metabolic profile of the offspring, these results are already published. Regarding Figure 9, we considered that this scheme summarizes the main results obtained in simplified form and the behavioral consequences induced by the model. All the information directly resulted from the results obtained and no further information was included. Indeed, the figure was acknowledged by the other reviewers.

Round 2

Reviewer 1 Report

Accept the revised version 

Reviewer 3 Report

The queries were adequately addressed